# UNIFORM DISCRETE DIFFUSION WITH METRIC PATH FOR VIDEO GENERATION

**Haoge Deng**[1,3,5*]**, Ting Pan**[2,3,5*]**, Fan Zhang**[5*]**, Yang Liu**[4,5*]**, Zhuoyan Luo**[5]**, Yufeng Cui**[5]
**Wenxuan Wang**[5]**, Chunhua Shen**[4]**, Shiguang Shan**[2,3]**, Zhaoxiang Zhang**[1,3†]**, Xinlong Wang**[5†]

[1]National Laboratory of Pattern Recognition, CASIA
[2]Key Laboratory of Intelligent Information Processing, ICT, CAS
[3]University of Chinese Academy of Sciences
[4]Zhejiang University     [5]Beijing Academy of Artificial Intelligence

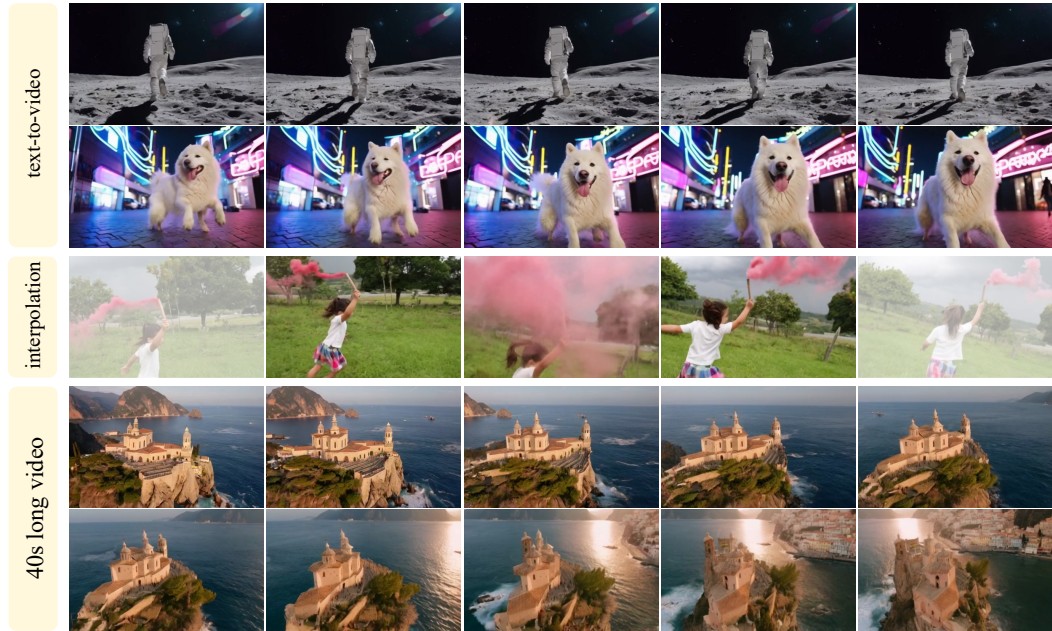

Figure 1: Visualization of **URSA** across diverse video generation tasks: text-to-video generation, video interpolation, and long video generation. These examples underscore the versatility of URSA.

## ABSTRACT

Continuous-space video generation has advanced rapidly, while discrete approaches lag behind due to error accumulation and long-context inconsistency. In this work, we revisit discrete generative modeling and present **U**niform disc**R**ete diffu**S**ion with metric p**A**th (**URSA**), a simple yet powerful framework that bridges the gap with continuous approaches for the scalable video generation. At its core, URSA formulates the video generation task as an iterative global refinement of discrete spatiotemporal tokens. It integrates two key designs: a Linearized Metric Path and a Resolution-dependent Timestep Shifting mechanism. These designs enable URSA to scale efficiently to high-resolution image synthesis and long-duration video generation, while requiring significantly fewer inference steps. Additionally, we introduce an asynchronous temporal fine-tuning strategy that unifies versatile tasks within a single model, including interpolation and image-to-video generation. Extensive experiments on challenging video and image generation benchmarks demonstrate that URSA consistently outperforms existing discrete methods and achieves performance comparable to state-of-the-art continuous diffusion methods. Code and models are available at https://github.com/baaivision/URSA

---

*Equal Contribution. This work was done when H.Deng, T.Pan, and Y.Liu were interns at BAAI.
†Corresponding Author: *wangxinlong@baai.ac.cn*, *zhaoxiang.zhang@ia.ac.cn*

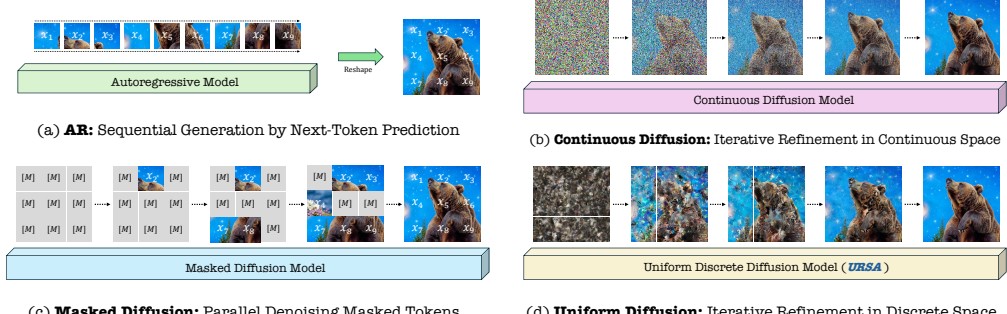

(a) **AR:** Sequential Generation by Next-Token Prediction

(b) **Continuous Diffusion:** Iterative Refinement in Continuous Space

(c) **Masked Diffusion:** Parallel Denoising Masked Tokens

(d) **Uniform Diffusion:** Iterative Refinement in Discrete Space

Figure 2: **Illustration of different image/video generation paradigms.** Discrete-space approaches such as AR and MDM adopt non-refinable local generation, where produced tokens are fixed once generated. In contrast, URSA introduces iterative global refinement, conceptually aligning discrete methods with continuous-space approaches, and substantially narrowing their performance gap.

# 1 INTRODUCTION

Continuous-space visual generation has achieved remarkable progress in both image and video synthesis (Batifol et al., 2025; Baldridge et al., 2024; Betker et al., 2023; Brooks et al., 2024; Wang et al., 2025a; Gao et al., 2025b; Yang et al., 2025b; Kong et al., 2024). Driven by advances in diffusion model algorithms (Ho et al., 2020; Song et al., 2021), these continuous-space methods have demonstrated strong capabilities in producing high-fidelity and visually coherent content, establishing themselves as the dominant paradigm for generative modeling.

In parallel, discrete-space text generation has become the *de facto* paradigm for large language models (Radford et al., 2018; 2019; Brown et al., 2020). Inspired by the success of LLMs, recent works have extended similar ideas to visual generation through discrete tokenization, using either next-token prediction (Sun et al., 2024a; Wang et al., 2024b; Kondratyuk et al., 2024) or masked token prediction (Chang et al., 2023; Xie et al., 2025c). However, discrete approaches still lag behind their continuous counterparts, facing challenges such as error accumulation and maintaining long-context consistency, especially in video generation. For instance, even though masked diffusion models employ bidirectional transformers, we still observe low visual quality and unnatural object motions.

In this work, we first revisit discrete generative modeling and introduce **URSA**, a powerful visual generation framework built upon **U**niform disc**R**ete diffu**S**ion with metric p**A**th. Our approach is simple: we generate videos and images by iterative refinement over discrete spatiotemporal tokens. As illustrated in Fig. 2, unlike classic autoregressive (AR) models and masked diffusion models (MDM) that adopt non-refinable local generation, where produced tokens are fixed once generated, URSA emphasizes *iterative refinement over global discrete tokens*, conceptually aligning discrete methods with continuous counterparts, and substantially narrowing their performance gap. URSA starts from categorical noise, $x_0 \sim \mathrm{Unif}([K])^D$, where each of the $D$-dimensional discrete token is independently sampled from a uniform distribution over the vocabulary $[K] = \{1, 2, \ldots, K\}$, and iteratively performs global refinement along a metric-guided probability path to obtain $x_1$ on the data manifold, *i.e.*, the target image or video. This iterative process enables URSA to capture the hierarchical structure of video data, from global layouts to detailed dynamics, while leveraging temporal redundancy to preserve spatiotemporal coherence.

We propose a novel metric probability path tailored for long visual sequences by incorporating two key components: a linearized metric path and a resolution-dependent timestep shifting mechanism. Collectively, these designs enable precise control over data perturbations, a property essential for effectively learning hierarchical data manifolds. This construction allows URSA to scale efficiently to long-sequence tasks, such as high-resolution image synthesis and long video generation, while requiring substantially fewer inference steps. Furthermore, we introduce an asynchronous timestep scheduling strategy, where timesteps are independently sampled for each frame. This asynchronous design empowers URSA to generate minute-level long videos and support a wide range of tasks within a unified model, including image-to-video generation, video interpolation, and extrapolation.

URSA achieves a text-to-video score of 82.4 on VBench (Huang et al., 2024a), outperforming discrete and continuous baselines. In image-to-video generation tasks, URSA reaches a VBench score of 86.2,

on par with the state-of-the-art open-source models. For text-to-image generation, URSA achieves a DPG-Bench (Hu et al., 2024) score of 86.0, exceeding previous discrete approaches. Moreover, URSA exhibits strong zero-shot generalization in variable length contexts, underscoring its versatility.

Our contributions can be summarized as follows: 1) We propose **URSA**, a simple yet powerful framework that bridges the gap to continuous diffusion methods and enables scalable video generation. 2) We highlight two key designs, Linearized Metric Path and Resolution-dependent Timestep Shifting for long-sequence training. We further propose an asynchronous timestep scheduling strategy that enables multi-task video generation. 3) URSA substantially pushes the envelope of discrete generation, attaining state-of-the-art performance on VBench, DPG-Bench, and GenEval (Ghosh et al., 2024).

## 2 RELATED WORKS

### 2.1 CONTINUOUS-SPACE VISUAL GENERATION

Continuous methods for visual generation have achieved significant progress in recent years. Early endeavors such as variational autoencoders (VAEs) (Kingma & Welling, 2014) and flow-based models (Dinh et al., 2014; 2017) exploit continuous latent spaces to model complex images, while GANs (Goodfellow et al., 2014) generate high-resolution images with strong perceptual quality via adversarial training (Brock et al., 2019; Karras et al., 2020). Diffusion models (Ho et al., 2020; Song et al., 2021), which learn to recover data by progressively denoising Gaussian noise in a continuous space, demonstrated remarkable performance in both image and video generation (Gao et al., 2025a; Batifol et al., 2025; Baldridge et al., 2024; Betker et al., 2023; Wu et al., 2025a; Brooks et al., 2024; Kong et al., 2024; Gao et al., 2025b; Wang et al., 2025a; Kuaishou, 2024; Ma et al., 2025). MAR (Li et al., 2024) employs an autoregressive framework with a diffusion head to produce continuous-valued outputs, and NOVA (Deng et al., 2025b) further extends this idea to video generation, applying autoregressive modeling to spatiotemporal sequences. URSA shares the same spirit as continuous diffusion models, performing global iterative refinement, but operates over discrete tokens.

### 2.2 DISCRETE-SPACE VISUAL GENERATION

Discrete visual generation can be broadly categorized into autoregressive and masked diffusion models, both operating on discrete visual tokens such as pixels (Kalchbrenner et al., 2017; Reed et al., 2017) or latent codes (Oord et al., 2017; Esser et al., 2021). Autoregressive models generate discrete visual tokens sequentially, with each prediction conditioned on previously generated context. This approach has been applied to both image (Sun et al., 2024a; Ramesh et al., 2021; Ding et al., 2021; Yu et al., 2022; Zhu et al., 2025) and video synthesis (Wang et al., 2024b; Yan et al., 2021; Kondratyuk et al., 2024; Wang et al., 2024c). Although simple in concept, this design often has slow inference and significant error accumulation. In contrast to autoregressive methods, masked diffusion models (Gat et al., 2024; Chang et al., 2022; 2023; Yu et al., 2023) introduce the prediction of masked tokens, enabling parallel generation and improved modeling of global context. Despite these advantages, it remains challenging to apply these methods to long sequences, *e.g.* high-fidelity long-form video. FUDOKI (Wang et al., 2025b) investigates the integration of discrete flow matching (Gat et al., 2024) within native multimodal models. In this work, we adopt a uniform discrete diffusion approach, which performs iterative global refinement from categorical noise. By addressing key challenges, URSA enables both efficient inference and high-quality long-sequence generation.

## 3 METHODOLOGY

We first review the concepts of uniform discrete diffusion / discrete flow matching in Sec. 3.1, which provide the theoretical foundation for our framework. In Sec. 3.2.1-3.2.2, we introduce URSA, a simple yet powerful framework that bridges the gap between discrete and continuous approaches, enabling effective and scalable video generation.

### 3.1 PRELIMINARY: DISCRETE FLOW MATCHING

Discrete Flow Matching (DFM) (Gat et al., 2024; Shaul et al., 2025) introduces a family of generative models designed to map data from an initial distribution $p_0(x)$, to a final distribution $p_1(x)$, within a

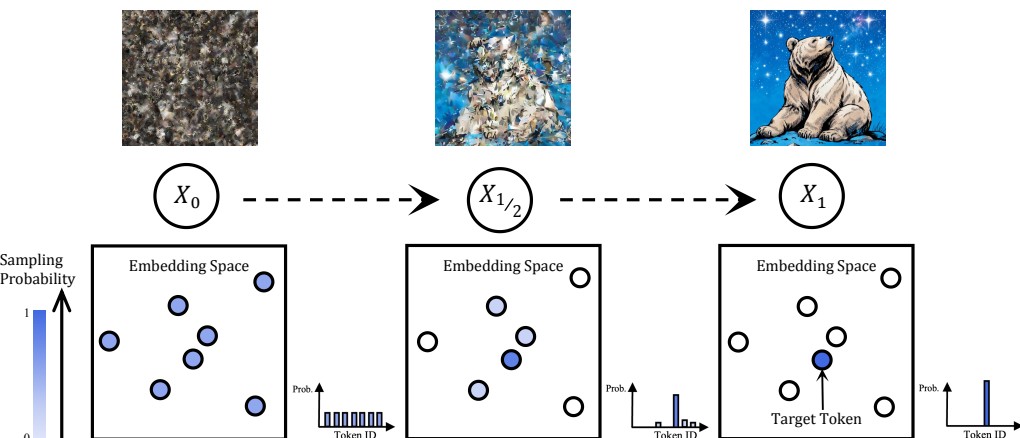

Figure 3: **Global refinement via token distance in embedding space.** Starting from categorical noise $x_0$ (left), our framework refines data based on token distance to get target data $x_1$ (right), enabling hierarchical structure generation from global semantics to fine details.

discrete state space. The model utilizes a time-dependent probability path, $p_t(x)$, which interpolates between these two distributions over the interval $t \in [0, 1]$. The key idea behind DFM is to define a velocity field, $u_t$, which drives the evolution of this probability path, enabling the model to simulate a Markov process and generate new data samples.

**Probability paths.** We consider the probability path $p_t(x)$, where $t \in [0, 1]$ indexes a time-dependent probability distribution between a source distribution $p_0(x)$ and a target distribution $p_1(x)$ over $t$. Given a data distribution $q(x)$ over $x = (x^1, \ldots, x^D) \in [K]^D$, the probability path is defined as

$$p_t(x) \triangleq \sum_{x_1 \in \mathcal{S}} p_t(x|x_1) q(x_1), \text{ where } p_t(x|x_1) \triangleq \prod_{i=1}^{D} p_t(x^i|x_1^i), \tag{1}$$

$p_t(x^i \mid x_1^i)$ denotes a *conditional* forward probability path, characterizing the evolution of the state $x^i$ given the initial state $x_1^i$.

**Probability velocities.** To generate the predefined probability path $p_t(x)$, we consider a Continuous-Time Markov Chain (CTMC), modeled as a stochastic process. The dynamics of this CTMC are governed by a probability velocity $u_t$, also known as the *transition rate*. The transition rate models how the current state $x_t$ evolves toward the target state $x_1$ over time. Within this framework, each token $i$ is updated independently according to the following transition rule:

$$x_{t+h}^i \sim \delta_{x_t^i}(\cdot) + h\, u_t^i(\cdot \mid x_t^i, x_1^i), \tag{2}$$

where $u_t^i(\cdot \mid x_t^i, x_1^i)$ represents *velocity field*, a conditional rate function that governs the flow of probability from the current state $x_t^i$ to the target state $x_1^i$ over time. Equation (2) can be interpreted as a small perturbation of the point mass $\delta_{x_t^i}$, scaled by the step size $h$, effectively modeling discrete state transitions as a continuous-time stochastic process. This velocity field is central to DFM, as it characterizes the dynamics of the probability path and is the primary quantity learned during training.

## 3.2 Uniform Discrete Diffusion with Metric Path

We present URSA, a novel framework built on uniform discrete diffusion with a metric path for image and video generation. In this section, we first introduce three key innovations: (1) a Linearized Metric-Path for structured and tractable trajectory design, (2) a Resolution-dependent Timestep Shifting mechanism to improve training stability and representation learning for long video sequences, and (3) a Frame-wise Independent Perturbation Scheduling strategy for unified long-video generation and multitask learning. After introducing these core components, we further provide the training procedure and sampling algorithm.

### 3.2.1 METRIC PROBABILITY PATH FOR LONG SEQUENCE DATA

For data with varying sequence lengths, the degree of perturbation should be adapted during training. This requires a probability path to effectively handle sequences of different lengths, such as high-resolution images or videos. In this section, we introduce two key techniques, linearized metric path and resolution-dependent timestep shifting, to address this challenge, ensuring that the perturbation process is appropriately adjusted based on the sequence length.

**Linearized metric path.** Inspired by (Shaul et al., 2025), We introduce the linearized metric path, a novel probability path derived from token embedding distances. Formally, we define the distance function $d : \mathcal{T} \times \mathcal{T} \to \mathbb{R}_{\geq 0}$, which measures the discrepancy between the codebook embeddings of generated token $x$ and the target tokens $x_1$. The distance satisfies the property $d(x, x_1) = 0 \Leftrightarrow x = x_1$, ensuring a well-defined metric structure. Based on this, the probability path is defined as

$$p_t(x|x_1) = \text{softmax}\left(-\beta_t d(x, x_1)\right) = \prod_{i=0} \text{softmax}\left(-\beta_t d(x^i, x_1^i)\right), \tag{3}$$

where $\beta_t : [0, 1] \to \mathbb{R}_{\geq 0}$ is a monotonic scheduler function with boundary conditions $\beta_0 = 0$, $\beta_1 = \infty$. The core of linearized path lies in the functional form of $\beta_t$, which is parameterized as

$$\beta_t = c \times \left(\frac{t}{1-t}\right)^\alpha, t \in [0, 1), \tag{4}$$

where $c > 0$ and $\alpha > 0$ are hyperparameters that control the relationship between the sampling distance $d(x_t, x_1)$ and time $t$. Specifically, the forward process samples $x_t \sim p_{t|1}(\cdot \mid x_1)$, with boundary conditions yielding a uniform distribution over codebook embeddings at $t = 0$ and a deterministic sample at $x_1$ when $t = 1$, illustrated in Figure 3. When $t$ is between 0 and 1, our objective is to find an appropriate set of values for $c$ and $\alpha$ that preserve the linear relationship between $t$ and $d(x_t, x_1)$. This linearity provides a finer control of perturbations over the probability path, as described next. Additional experiments and discussions on the impact of linearized metric path on model convergence speed and overall performance are provided in Sec. 4.3.

**Resolution-dependent timestep shifting.** Intuitively, since higher resolutions contain more pixels, more perturbation is needed to alter the signal. To address this, we introduce a time shift parameter $\lambda$, which adjusts the timestep based on the resolution. For any given $t$, we define the shifted timestep $\tilde{t}$

$$\tilde{t} = \frac{t}{t + \lambda(1-t)}. \tag{5}$$

Because our proposed linearized metric path enforces a linear relationship between $t$ and $d(x_t, x_1)$, we modulate this path using $\lambda$ to accommodate varying data resolutions. For higher resolutions, we set $\lambda > 1$ to create a convex relationship between $\tilde{t}$ and $d(x_t, x_1)$ that introduces stronger perturbations. For lower resolutions, we set $\lambda < 1$, yielding a concave relationship with more gradual perturbations.

### 3.2.2 ASYNCHRONOUS TIMESTEP SCHEDULING

Due to their complex spatiotemporal dynamics and broad applicability across downstream tasks, prior video generation methods render task-specific modeling both inefficient and resource-intensive. Motivated by diffusion forcing (Chen et al., 2024a), we propose an asynchronous timestep scheduling strategy tailored for multi-task training and sampling. Rather than applying the same noise level across all frames in a video sequence (*i.e.*, synchronous timestep scheduling), we assign uniform noise levels independently to each frame. Formally, given a clean video sequence $\mathbf{F} = \{f^{(1)}, f^{(2)}, \ldots, f^{(n)}\}$ with $n$ frames, we assign each frame a continuous time $t_i \sim \mathcal{U}(0, 1)$, forming the timestep schedule $\mathbf{T} = \{t_1, t_2, \ldots, t_n\}$. The corresponding noisy sequence is denoted as $\tilde{\mathbf{F}} = \{f_{t_1}^{(1)}, f_{t_2}^{(2)}, \ldots, f_{t_n}^{(n)}\}$, where the metric-induced probability path in Eq. 3 is applied frame-wise according to its assigned $t_i$. This strategy enables fine-grained temporal modeling over the timestep schedule via decoupling noise levels across frames. As a result, the training progress adaptively balances local frame reconstruction with global temporal coherence, facilitating versatile generation objectives, such as text-to-video, image-to-video, video extrapolation, and start–end frame control within a unified model architecture. Additional visualizations for these advanced generation tasks are provided in Appendix C & D.

### 3.2.3 TRAINING AND SAMPLING

**Training.** We first encode video clips into discrete token sequences using a pre-trained tokenizer, resulting in a clean video sequence $x_1 = \{f_1^{(1)}, f_1^{(2)}, \ldots, f_1^{(n)}\}$, where $n$ denotes the number of video frames and $f_1^{(i)}$ denotes the $i$-th frame tokens. At each training step, we uniformly sample timesteps $t_i \in [0, 1]$ for each frame $f_1^{(i)}$ in the sequence and obtain a perturbed sequence $x_t \sim p_t(\cdot \mid x_1)$ via the proposed metric probability path. The backbone, implemented with the LLM architecture, takes as input the concatenation of text tokens $e$ and noisy tokens $x_t$, and produces logits over the token vocabulary to predict the original sequence $x_1$. The training objective is formulated as the expected cross-entropy between the ground-truth visual tokens and the model's predicted distribution:

$$\mathcal{L} = \mathbb{E}_{t \sim \mathcal{U}[0,1],\, x_1, x_t} \left[ -\log p_{1|t}(x_1 \mid x_t, e) \right]. \tag{6}$$

**Sampling.** We follow Gat et al. (2024); Shaul et al. (2025) and employ the Euler solver for efficient and high-quality generation. Specifically, we first uniformly sample $x_0$ from the full vision vocabulary and feed it into the model to obtain the prediction $\hat{x}_1$. Following Shaul et al. (2025), we compute the velocity field $u_t(\cdot \mid x_t, \hat{x}_1)$. We then iteratively refine $x_t$ using $u_t$, where each iteration updates the sample $x_t$ along the estimated direction. Once the $T$ refinement steps have been completed, the sampling process returns a clean image or video. Additional details are provided in Appendix A.

## 4 EXPERIMENT

### 4.1 EXPERIMENT SETUP

**Datasets.** We leverage a curated selection of high-quality datasets to effectively train URSA models. For text-to-image training, we collect 16M image-text pairs sourced from Unsplash (Unsplash, 2020), DataComp (Gadre et al., 2024), COYO (Byeon et al., 2022), and JourneyDB (Sun et al., 2023). These pairs are filtered by image resolution and aesthetic score, and further supplemented with 14M AI-generated image samples using the FLUX.1 model (Batifol et al., 2025). For text-to-video training, we select 12M video-text pairs from the highest scoring subset of Koala-36M (Wang et al., 2025c) and complement them with 12M internal video-text pairs. The internal videos are captioned using the Emu2-17B model (Sun et al., 2024b) in conjunction with the captioning engine (Diao et al., 2024). We uniformly sample short and long captions during training, with a maximum length of 320 tokens.

**Architectures.** We initialize our visual generation model with weights from a pre-trained LLM. Specifically, we adopt the Qwen3 LLM architecture (Yang et al., 2025a), which natively incorporates QK-Norm (Dehghani et al., 2023) layer to stabilize the multimodal training. To better capture the spatiotemporal structure inherent in videos, we introduce an enhanced M-RoPE (Wang et al., 2024a) that allocates interleaved frequency components across temporal, height, and width dimensions, following the approach of Mogao (Liao et al., 2025). Crucially, unlike Liao et al. (2025), our 3D-RoPE assigns identical positions for texts, ensuring equivalence with the 1D-RoPE (Su et al., 2024). We use the Cosmos (Agarwal et al., 2025) tokenizer to extract image and video tokens, achieving $4\times$ temporal and $8\times8$ spatial compression through a 64K FSQ (Mentzer et al., 2024) codebook. Furthermore, we train an IBQ (Shi et al., 2025) tokenizer for high-resolution image generation, facilitating efficient $16\times16$ spatial compression via a 256-dimensional codebook with 131K entries.

**Diffusion schedulers.** We adopt the Kinetic Optimal Scheduler (Shaul et al., 2025), equipped with a metric-induced probability path specifically designed for the embedding space of vision tokenizers. Following Shaul et al. (2025), we perform a grid search over the path hyperparameters $\alpha$ and $c$, visually inspecting the reconstructed samples for each $(\alpha, c)$ that fully exploit the time interval $[0, 1]$. Eventually, we select $(\alpha, c)$ to $(1.0, 5)$ for the Cosmos tokenizer and $(0.5, 6)$ for our IBQ tokenizer. For standard uniform diffusion, we use the mixture probability path proposed by Gat et al. (2024). In contrast, for masked diffusion, we adopt the MaskGIT (Chang et al., 2022) scheduler, which has been empirically shown to achieve state-of-the-art performance in both image and video generation models (Kondratyuk et al., 2024; Bai et al., 2025). Following established practice in continuous diffusion models, we default to 25 inference steps for image generation and 50 for video generation.

**Training details.** URSA is trained on 128 A100 (40GB) GPUs. In all experiments, we use the AdamW optimizer (Loshchilov & Hutter, 2019) with $\beta_1 = 0.9$, $\beta_2 = 0.95$, weight decay of 0.05, and an initial learning rate of 1e-4. The learning rate employs cosine decay (Loshchilov & Hutter, 2017).

We first pre-train text-to-image models and leverage their weights to initialize text-to-video models. Subsequently, following Chen et al. (2025a), we adapt full-sequence video diffusion models to diffusion forcing architectures by applying frame-wise noise schedules for autoregressive generation.

**Evaluation.** We evaluate text-to-image alignment using benchmarks DPG-Bench (Hu et al., 2024) and GenEval (Ghosh et al., 2024). Each image is generated from original or rewritten text prompts, with resolution determined by model type: 1024×1024 for image generation models to support high fidelity, and 512×320 for video generation models to effectively measure cross-modal generalization. We access text-to-video generation using VBench (Huang et al., 2024a) and image-to-video generation with VBench++ (Huang et al., 2024b), its comprehensive successor tailored for real-world scenarios. The videos, sized 49×512×320, are generated from rewritten prompts for text-to-video evaluation, and from original text prompts with official cropped first-frame images for image-to-video evaluation. We apply classifier-free guidance (Ho & Salimans, 2022) with a scale value of 7.0 in all evaluations.

## 4.2 MAIN RESULTS

**URSA rivals Sora-like text-to-video generation models despite using a discrete video tokenizer.** Current discrete video tokenizers offer limited spatiotemporal compression and reconstruction quality, posing significant challenges to bidirectional diffusion transformers. However, URSA excels in generating video clips from text, achieving strong performance on the VBench, as shown in Table 1. Compared to Sora-like diffusion models: Vchitect (Fan et al., 2025), Pyramid Flow (Jin et al., 2025), LuminaVideo (Liu et al., 2025a), OpenSora (Peng et al., 2025) and OpenSoraPlan (Lin et al., 2024), URSA matches or exceeds their performance, particularly in the semantic field. These results further underscore the need for a tokenizer that satisfies the imaging quality of state-of-the-art continuous models (Kong et al., 2024; Teng et al., 2025; Ma et al., 2025; Yang et al., 2025b; Wang et al., 2025a).

Table 1: **Text-to-video evaluation on VBench.** For clarity and to better highlight distinctions between models, we report only the most relevant metrics across the quality and semantic dimensions.

| Model | #params | #videos | Total Score | Quality Score | Semantic Score | Dynamic Degree | Aesthetic Quality | Imaging Quality | Object Class | Multiple Objects | Spatial Relationship | Color | Scene |
|---|---|---|---|---|---|---|---|---|---|---|---|---|---|
| ▼ *Continuous models* | | | | | | | | | | | | | |
| NOVA | 0.6B | 20M | 80.1 | 80.4 | 79.1 | 20.1 | 59.4 | 59.4 | 92.0 | 77.5 | 77.5 | 87.7 | 54.1 |
| Vchitect-2.0 | 2B | 134M | 81.6 | 82.5 | 77.8 | 58.3 | 61.5 | 65.6 | 87.8 | 69.4 | 54.6 | 86.9 | 57.5 |
| Pyramid Flow | 2B | 10M | 81.7 | 84.7 | 69.6 | 64.6 | 63.3 | 65.0 | 86.7 | 50.7 | 59.5 | 82.9 | 43.2 |
| LuminaVideo | 2B | 12M | 83.0 | 83.9 | 79.3 | 67.1 | 62.3 | 64.6 | 91.0 | 68.3 | 67.3 | 90.2 | 56.1 |
| OpenSoraPlan v1.5 | 8B | 40M | 83.0 | 84.2 | 78.2 | 64.4 | 66.9 | 68.5 | 91.9 | 70.7 | 80.1 | 81.8 | 52.1 |
| OpenSora 2.0 | 11B | 85M | 83.6 | 84.4 | 80.3 | 56.4 | 65.3 | 65.7 | 94.6 | 78.0 | 76.8 | 86.3 | 53.4 |
| MAGI-1 | 24B | - | 81.8 | 84.7 | 70.4 | 72.5 | 59.3 | 65.3 | 84.1 | 50.6 | 73.0 | 87.5 | 28.9 |
| Step-Video | 30B | - | 81.8 | 84.5 | 71.3 | 53.1 | 61.2 | 70.6 | 80.6 | 50.6 | 71.5 | 88.3 | 24.4 |
| CogVideoX1.5 | 5B | - | 82.0 | 82.7 | 79.2 | 56.2 | 62.1 | 65.3 | 83.4 | 65.3 | 79.4 | 88.4 | 53.3 |
| HunyuanVideo | 13B | - | 83.2 | 85.1 | 75.8 | 70.8 | 60.4 | 67.6 | 86.1 | 68.6 | 68.7 | 91.6 | 53.9 |
| Wan2.1 | 14B | - | 83.7 | 85.6 | 76.1 | 65.5 | 66.1 | 69.4 | 86.3 | 69.6 | 75.4 | 88.6 | 45.8 |
| ▼ *Discrete models* | | | | | | | | | | | | | |
| Lumos-1 | 3.6B | 10M | 78.3 | 79.5 | 73.5 | - | - | 58.0 | 90.1 | - | - | 82.0 | - |
| Emu3 | 8B | - | 81.0 | 84.1 | 68.4 | 79.3 | 59.6 | 62.6 | 86.2 | 44.6 | 68.7 | 88.3 | 37.1 |
| URSA | 1.7B | 24M | 82.4 | 83.4 | 78.5 | 81.4 | 63.1 | 62.2 | 93.4 | 70.6 | 62.1 | 90.7 | 52.3 |

**URSA emerges frame-conditioned video generation by accurately modeling the future motion.** Prior methods typically adapt text-to-image (Ren et al., 2024; Chen et al., 2024b; Xing et al., 2024) or text-to-video models with a clean first frame for image-to-video generation. In contrast, URSA seamlessly integrates asynchronous frame conditions, enabling zero-shot generalization for this task. As depicted in Table 2, URSA excels in camera control and subject movement versus specialized frame-conditioned models (Agarwal et al., 2025; Yu et al., 2025; Wang et al., 2025a; Liu et al., 2025b). Our results demonstrate that diffusion forcing effectively generalizes to image-to-video generation, pushing the boundaries of autoregressive discrete video generation models without causal attention.

**URSA performs on par with the state-of-the-art models in generating high-resolution images.** We compare URSA against continuous models in Table 3, encompassing specialist architectures: SDXL (Podell et al., 2024), SD3 (Esser et al., 2024), FLUX (Batifol et al., 2025), SANA (Xie et al., 2025a) and NOVA (Deng et al., 2025b), as well as unified architectures: Mogao (Liao et al., 2025), Bagel (Deng et al., 2025a), OmniGen2 (Wu et al., 2025b) and Show-o2 (Xie et al., 2025d). Through joint modeling of discrete text and visual tokens, URSA demonstrates strong text-image alignment. For example, on the DPG-Bench, URSA reaches a leading overall score with dense text prompts. This strong performance is consistently sustained on the GenEval when using the rewritten prompts.

Table 2: **Image-to-video evaluation on VBench++.** To evaluate temporal consistency, we focus on image-to-video (I2V) metrics of visual similarity between each generated frame and reference image.

| Model | #params | #videos | Total Score | Quality Score | I2V Score | Dynamic Degree | Aesthetic Quality | Imaging Quality | Camera Motion | I2V Subject Consistency | I2V Background Consistency |
|---|---|---|---|---|---|---|---|---|---|---|---|
| ▼ *Continuous models* | | | | | | | | | | | |
| ConsistI2V | 2B | 10M | 84.1 | 76.2 | 91.9 | 18.6 | 59.0 | 66.9 | 33.9 | 95.8 | 96.0 |
| I2VGen-XL | 2B | 35M | 85.3 | 78.4 | 92.1 | 26.1 | 64.8 | 69.1 | 18.5 | 96.5 | 96.8 |
| SEINE | 3B | 25M | 85.5 | 78.4 | 92.7 | 27.1 | 64.6 | 71.4 | 21.0 | 97.2 | 97.0 |
| DynamiCrafter | 2B | 10M | 86.9 | 80.5 | 93.5 | 69.7 | 60.9 | 68.6 | 31.2 | 97.2 | 97.4 |
| Cosmos | 13B | 100M | 84.2 | 75.8 | 92.6 | 18.7 | 55.8 | 59.9 | 25.4 | 96.0 | 97.4 |
| VideoMAR | 1.4B | 0.5M | 84.8 | 75.6 | 94.0 | 11.0 | 55.8 | 62.3 | 21.6 | 97.9 | 98.4 |
| CogVideoX | 5B | - | 86.7 | 78.6 | 94.8 | 33.2 | 61.9 | 70.0 | 67.7 | 97.2 | 96.7 |
| HunyuanVideo | 13B | - | 86.8 | 78.5 | 95.1 | 22.2 | 62.6 | 70.1 | 49.9 | 98.5 | 97.4 |
| Wan2.1 | 14B | - | 86.9 | 80.8 | 92.9 | 51.4 | 64.8 | 70.4 | 34.8 | 97.0 | 96.4 |
| Pusa | 14B | - | 87.3 | 79.8 | 94.8 | 52.6 | 63.2 | 68.3 | 29.5 | 97.6 | 99.2 |
| Step-Video | 30B | - | 88.4 | 81.2 | 95.5 | 48.8 | 62.3 | 70.4 | 49.2 | 97.9 | 98.5 |
| MAGI-1 | 24B | - | 89.3 | 82.4 | 96.1 | 68.2 | 64.7 | 69.7 | 50.9 | 98.4 | 99.0 |
| ▼ *Discrete models* | | | | | | | | | | | |
| Lumos-1 | 3.6B | 10M | 84.7 | 76.1 | 93.3 | - | - | 69.2 | - | 97.4 | 97.4 |
| URSA | 1.7B | 24M | 86.2 | 79.8 | 92.6 | 65.3 | 57.4 | 64.2 | 37.6 | 96.1 | 96.5 |

At high resolutions, URSA surpasses the autoregressive (Wang et al., 2024b; Han et al., 2025; Chen et al., 2025b) and masked diffusion (Bai et al., 2025; Yuan et al., 2025) approaches in efficiency, effectively reducing inference steps through iterative refinement while preserving fine-grained detail.

Table 3: **Text-to-image evaluation on DPG-Bench and GenEval.** We prefer the DPG-Bench metrics to mitigate potential prompt template leakage concerns (Xie et al., 2025b) associated with GenEval. † refers to the methods using rewritten GenEval prompts for clearer position and attribute guidance.

| Model | ModelSpec | | DPG-Bench | | | | GenEval | | | | | | |
|---|---|---|---|---|---|---|---|---|---|---|---|---|---|
| | #params | #images | Overall | Entity | Attribute | Relation | Overall | Single | Two | Counting | Colors | Position | ColorAttr |
| ▼ *Continuous models* | | | | | | | | | | | | | |
| SDXL | 2.6B | - | 74.7 | 82.4 | 80.9 | 86.8 | 0.55 | 0.98 | 0.44 | 0.39 | 0.85 | 0.15 | 0.23 |
| SD3 | 2B | - | 84.1 | 91.0 | 88.8 | 80.7 | 0.62 | 0.98 | 0.74 | 0.63 | 0.67 | 0.34 | 0.36 |
| FLUX.1-dev | 12B | - | 84.9 | - | - | - | 0.68 | 0.99 | 0.85 | 0.74 | 0.79 | 0.21 | 0.48 |
| NOVA | 1.4B | 600M | 83.0 | 88.7 | 86.4 | 91.9 | 0.71 | 0.99 | 0.91 | 0.62 | 0.85 | 0.33 | 0.56 |
| SANA-1.5† | 4.8B | 50M | 84.7 | - | - | - | 0.81 | 0.99 | 0.93 | 0.86 | 0.84 | 0.59 | 0.65 |
| OmniGen2 | 7B | - | 83.6 | 88.8 | 90.2 | 89.4 | 0.80 | 1.00 | 0.95 | 0.64 | 0.88 | 0.55 | 0.76 |
| Mogao† | 7B | - | 84.3 | 90.0 | 88.3 | 93.2 | 0.89 | 1.00 | 0.97 | 0.83 | 0.93 | 0.84 | 0.80 |
| Bagel | 14B | - | 85.1 | 90.4 | 91.3 | 90.8 | 0.82 | 0.99 | 0.94 | 0.81 | 0.88 | 0.64 | 0.63 |
| Show-o2† | 7B | 66M | 86.1 | 91.8 | 90.0 | 91.8 | 0.76 | 1.00 | 0.87 | 0.58 | 0.92 | 0.52 | 0.62 |
| ▼ *Discrete models* | | | | | | | | | | | | | |
| Show-o | 1.3B | 2B | 67.3 | 75.4 | 78.0 | 84.5 | 0.68 | 0.98 | 0.80 | 0.66 | 0.84 | 0.31 | 0.50 |
| Emu3† | 8B | - | 81.6 | 87.2 | 86.3 | 90.6 | 0.66 | 0.99 | 0.81 | 0.42 | 0.80 | 0.49 | 0.45 |
| FUDOKI | 1.5B | 13M | 83.6 | 89.7 | 88.1 | 93.7 | 0.77 | 0.96 | 0.85 | 0.56 | 0.88 | 0.68 | 0.67 |
| Janus-Pro | 7B | 72M | 84.2 | 88.9 | 89.4 | 89.3 | 0.80 | 0.99 | 0.89 | 0.59 | 0.90 | 0.79 | 0.66 |
| Meissonic | 1B | 210M | - | - | - | - | 0.54 | 0.99 | 0.66 | 0.42 | 0.86 | 0.10 | 0.22 |
| Lumos-1† | 3.6B | 60M | - | - | - | - | 0.66 | 0.95 | 0.80 | 0.46 | 0.81 | 0.48 | 0.48 |
| Infinity† | 2B | - | 83.5 | - | - | 90.8 | 0.73 | 0.99 | 0.85 | 0.64 | 0.84 | 0.49 | 0.57 |
| URSA (512×320) | 1.7B | 30M | 82.5 | 88.3 | 86.4 | 92.9 | 0.64 | 0.99 | 0.83 | 0.47 | 0.83 | 0.30 | 0.41 |
| URSA (1024×1024) | 1.7B | 30M | 86.0 | 91.5 | 89.6 | 94.7 | 0.68 | 0.99 | 0.92 | 0.63 | 0.86 | 0.25 | 0.40 |
| URSA† (1024×1024) | 1.7B | 30M | - | - | - | - | 0.80 | 1.00 | 0.92 | 0.64 | 0.89 | 0.67 | 0.69 |

## 4.3 ABLATION STUDY

**Effectiveness of iterative refinement for visual generation.** Discrete diffusion models inherently incur elevated sampling errors, as exhibited in prior studies (Tang et al., 2022; Feng et al., 2025). To systematically investigate this issue in image and video generation, we train three variants of the discrete diffusion model, assessing performance across insufficient and excessive sampling regimes. Figure 4 compares key performance metrics of text-to-image models on GenEval and text-to-video models on VBench, with all models evaluated after being trained for an identical number of iterations. In the image generation task, which is characterized by low structural redundancy, all three models can generate feasible images within the conventional 25 inference steps. Without iterative refinement, reducing the number of steps substantially decreases the GenEval score in masked diffusion sampling. As we progress into video generation, a task rich in contextual redundancy, it becomes essential to correct sampling errors at each step, ensuring temporal coherence and visual fidelity across frames.

**Effectiveness of path linearity for uniform diffusion.** As shown in Figure 5, the left plot shows the average Euclidean distance between the embedding of noisy images and the clean image using

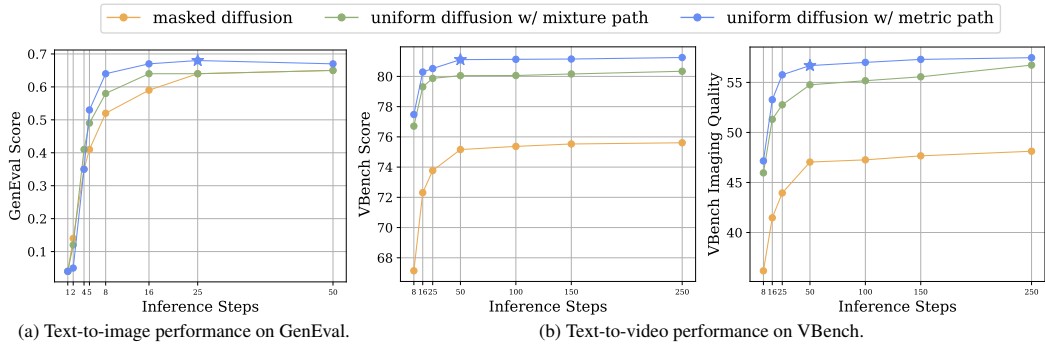

(a) Text-to-image performance on GenEval.

(b) Text-to-video performance on VBench.

Figure 4: **Sampling performance across inference steps.** Using the Cosmos tokenizer, we evaluate the image samples at 256×256 (∼1K tokens) and the video samples at 25×384×240 (∼10K tokens).

10K images sampled from the training set. We compute the Pearson correlation coefficients between the Euclidean distance and the timestep, which are -0.995, -0.921, -0.997, and -0.949. We find that the choice of the probability path is significantly influenced by the values of $c$ and $\alpha$, which has a substantial impact on the model performance. To determine the optimal values for $c$ and $\alpha$, we draw inspiration from the continuous diffusion model SD3 (Esser et al., 2024), where the relationship between $t$ and $d(x_t, x_1)$ demonstrates a strong linear correlation. This insight guides our approach to calibrating $c$ and $\alpha$ to effectively reach the limits of model performance for different vision tokenizers.

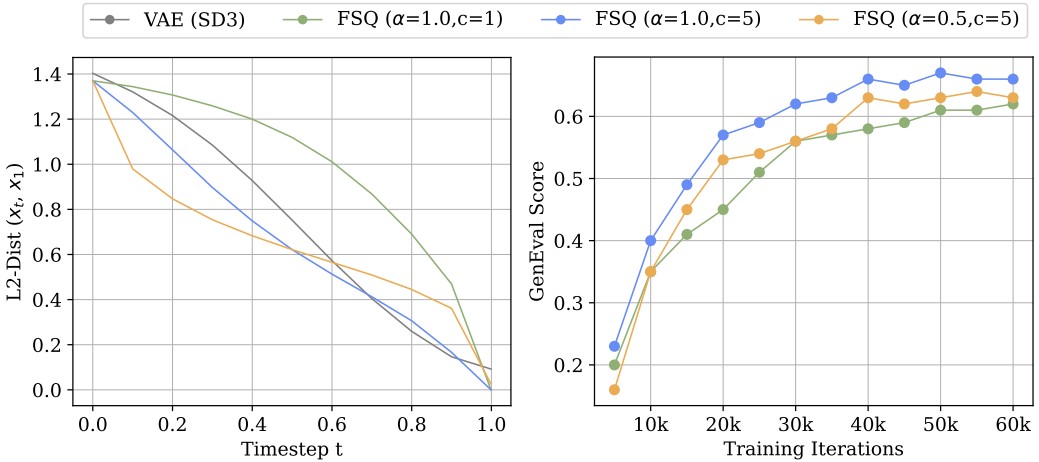

Figure 5: **Sampling performance of different paths.** We evaluate the image samples at 256×256.

**Effectiveness of model size for uniform diffusion.** To study the scaling properties of URSA models, we train three variants that are initialized from Qwen3 models with 0.6B, 1.7B, and 4B parameters. Figure 6 compares the performance of different model sizes on DPG-Bench, GenEval, and VBench, with all models trained for the same epoch count as in Sec. 4.2. We find that increasing model size considerably enhances semantic performance across both text-to-image and text-to-video evaluations but does not significantly improve generation quality. This suggests that while larger models better capture high-level semantics and align more accurately with text prompts, the fidelity of the generated outputs may ultimately be constrained by the representation capacity of the discrete vision tokenizer.

**Effectiveness of timestep shifting for video generation.** As outlined in Section 4.1, our probability path is designed to maximize the time interval. In line with continuous models (Kong et al., 2024; Wang et al., 2025a; Liu et al., 2025a), the optimal SNR schedule should be tailored with video size. To study the impact of the SNR schedule on video generation, we train four text-to-video models with divergent timestep shifting and evaluate their performance using the respective value on VBench. Figure 7 presents our shifting schedules, accompanied by their evaluation metrics and visualizations.

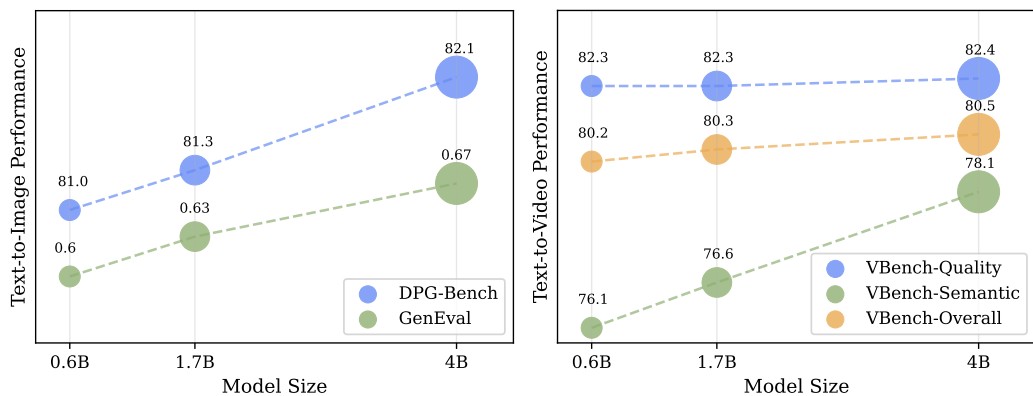

Figure 6: **Sampling performance of different model sizes.** All models are trained for the same epoch count as in the main experiments and evaluated on 256×256 images and 25×384×240 videos.

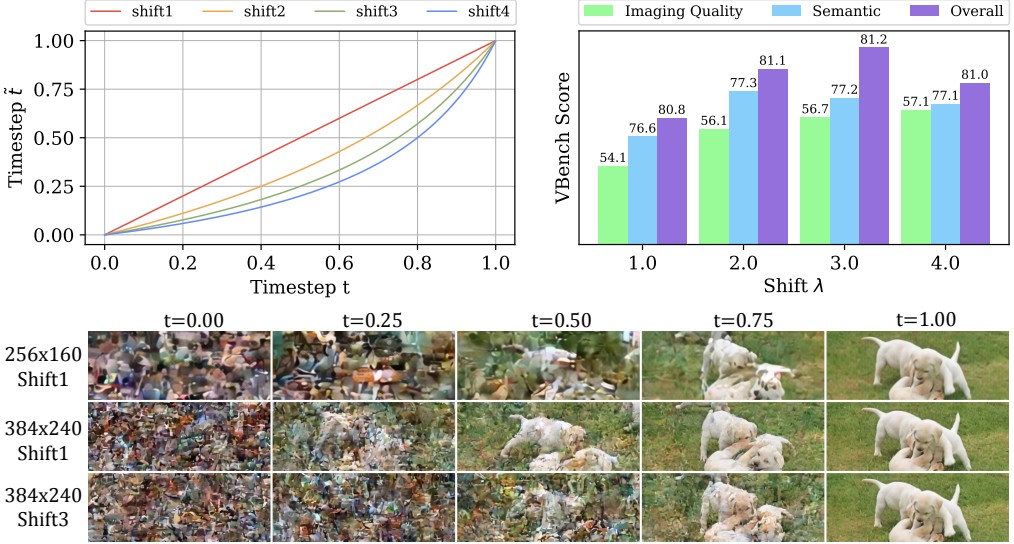

Figure 7: **Timestep shifting across SNR schedules.** We sample 25×384×240 videos for evaluation.

Surprisingly, the shifting strategy proposed by Esser et al. (2024) demonstrates strong effectiveness for uniform diffusion, empowering URSA to match the performance of its continuous counterparts.

## 5 CONCLUSION

In this work, we revisited discrete generative modeling for video synthesis and introduced **URSA**, a uniform diffusion framework with a metric path that bridges discrete and continuous paradigms. URSA employs two key innovations, a linearized metric path and a resolution-dependent timestep shifting strategy, to provide fine-grained control over perturbations for long sequences. On top of this, our asynchronous temporal scheduling strategy enables multi-task video generation in one model. Extensive experiments show that URSA not only consistently outperforms existing discrete methods but also achieves highly competitive results compared to state-of-the-art continuous diffusion models. We contend that this work represents a significant step toward unifying discrete and continuous paradigms and provides a promising direction for scalable, versatile, and efficient video generation.

**Acknowledgements.** This work was supported by the NSFC (No. 62320106010, U21B2042). We thank Jin Wang, Jiahao Wang, Zhengxiong Luo, Yuanzhi Zhu, Shilin Lu, Honghao Chen, Jinming Wu, Chengyuan Wang, and colleagues at BAAI for their valuable discussions and feedback on implementation and manuscript preparation.

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
