## APPENDIX

We have published code and pre-trained models to improve interpretability and ensure reproducibility. In this appendix, implementation details, experiments, and qualitative results are organized as follows:

- Training and Sampling Details (Sec. A)

- Additional Ablation Studies (Sec. B)

- Video extrapolation experiments (Sec. C)

- Start-End frame control experiments (Sec. D)

## A  TRAINING AND SAMPLING DETAILS

---

**Algorithm 1 URSA Training**

**Require:** Predictor $p_\theta$, Shift $\lambda$
1: **repeat**
2:      $x_1 \sim p_{\text{data}}$
3:      $t \sim \mathcal{U}(0, 1)$
4:      $\tilde{t} \leftarrow t/(t + \lambda(1 - t))$        ▷ time shift
5:      $x_{\tilde{t}} \sim p_{\tilde{t}|1}(\cdot|x_1)$
6:      $\mathcal{L} \leftarrow -\sum_{i=1}^{D} \log p_\theta(x_1^i \mid x_{\tilde{t}})$
7:      $\theta \leftarrow \theta - \eta \nabla_\theta \mathcal{L}$
8: **until** converged
9: **return** Trained predictor $p_\theta$

---

**Algorithm 2 URSA Sampling**

**Require:** Predictor $p_\theta$, Steps $T$, Shift $\lambda$, Visual vocabulary $\mathcal{V}$
1: Sample $x_0 \sim \text{Uniform}(\mathcal{V})$
2: **for** $k = 1$ to $T$ **do**
3:      $t \leftarrow (k - 1)/T$
4:      $\tilde{t} \leftarrow t/(t + \lambda(1 - t))$        ▷ time shift
5:      $\hat{x}_1 \sim p_\theta(\cdot|x_{\tilde{t}})$
6:      $u_{\tilde{t}} \leftarrow u_t(x, z|\hat{x}_1)$
7:      $x_{\tilde{t}} \leftarrow x_{\tilde{t}} + h \cdot u_{\tilde{t}}$
8: **end for**
9: **return** $x_1$        ▷ Generated discrete sample

---

**Training.** URSA is a predictor, a parametric model $p_\theta(\cdot|x_t)$, that takes $x_t$ as input and refine all tokens simultaneously. We first encode images and videos into discrete latent tokens using a pre-trained tokenizer. For visual tokens, we adopt a DFM training objective based on the probability path. At each iteration, we randomly sample a timestep $t \in [0, 1]$ and use the metric path to obtain the noised tokens $x_t$. Text prompts are tokenized using the Qwen3 tokenizer and embedded into the same semantic space. We concatenate text embeddings and noised visual tokens into a unified sequence. The training objective is defined as the expected cross-entropy between the ground-truth visual token sequence and the model's predicted distribution. We list the detailed training process in Algorithm 1.

**Sampling.** This velocity field ensures that transitions occur only from state $z$ to state $x$ when $x$ is closer to $x_1$ than $z$, *i.e.*, $d(x, x_1) < d(z, x_1)$. Using the distance metric and the time-dependent factor $\beta_t$, the velocity guides the flow of particles in a manner that is both kinetic-optimal and aligned with the underlying geometry of the state space. We list the complete sampling process in Algorithm 2.

## B  ADDITIONAL ABLATION STUDIES

### B.1  EFFECTIVENESS OF TIMESTEP CONDITIONING FOR UNIFORM DIFFUSION

Recent work explores time-agnostic (i.e., noise-unconditional) methods for both continuous diffusion (Sun et al., 2025; Tang et al., 2025) and masked diffusion (Zheng et al., 2025; Ou et al., 2025), effectively narrowing the architectural gap between diffusion transformers (DiTs) and LLMs. In this context, we analyze whether timestep conditioning remains indispensable for uniform diffusion. The results are illustrated in Figure 8. Specifically, we train three model variants with distinct conditioning strategies and evaluate GenEval across training iterations. After one epoch ($\sim$30K iterations), embedding or prompting with the timestep provides no measurable benefit. Notably, timestep embedding could degrade performance as their variance increases, potentially disrupting token embedding and compromising training stability.

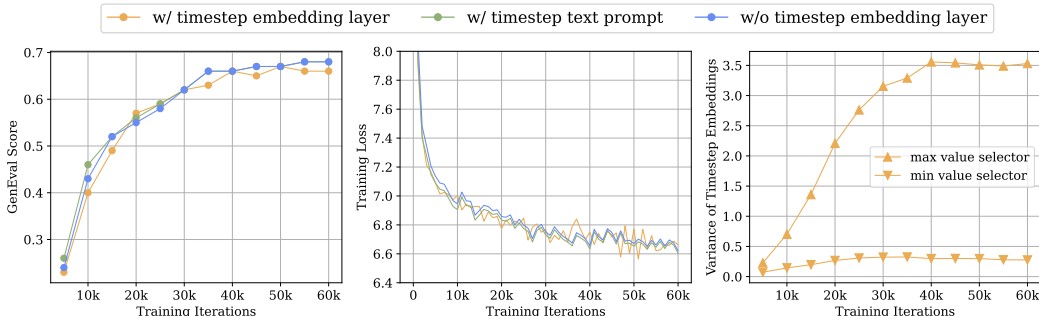

Figure 8: **Model metrics across training iterations.** We sample 256×256 images for evaluation.

## C  VIDEO EXTRAPOLATION EXPERIMENTS

As URSA is trained by applying independent noise levels to each frame, it naturally lends itself to video extrapolation via a sliding window. Specifically, new frames are generated sequentially, conditioned on the most recent 13 frames, thereby extending future predictions beyond the initial 49-frame context window. To effectively mitigate sampling errors in autoregressive video generation, we introduce a small amount of noise into historical frames by resampling them at timestep $t = 0.9$. Figure 9 presents the qualitative results for a video of 481 frames, where the initial text-to-video segment is extended through 12 extrapolation steps, producing videos up to $10\times$ the original length.

## D  START-END FRAME CONTROL EXPERIMENTS

We evaluate URSA on the start-end frame control task, a specialized form of video generation to prevent future predictions from drifting. Concretely, we extract a sequence of frames from the video at 4-second intervals and place them sequentially at the beginning and the end of the context window. This setup enables the generation of a video featuring coherent motion of both objects and cameras, preserving spatial relationships throughout the scene. We present the qualitative results in Figure 10.

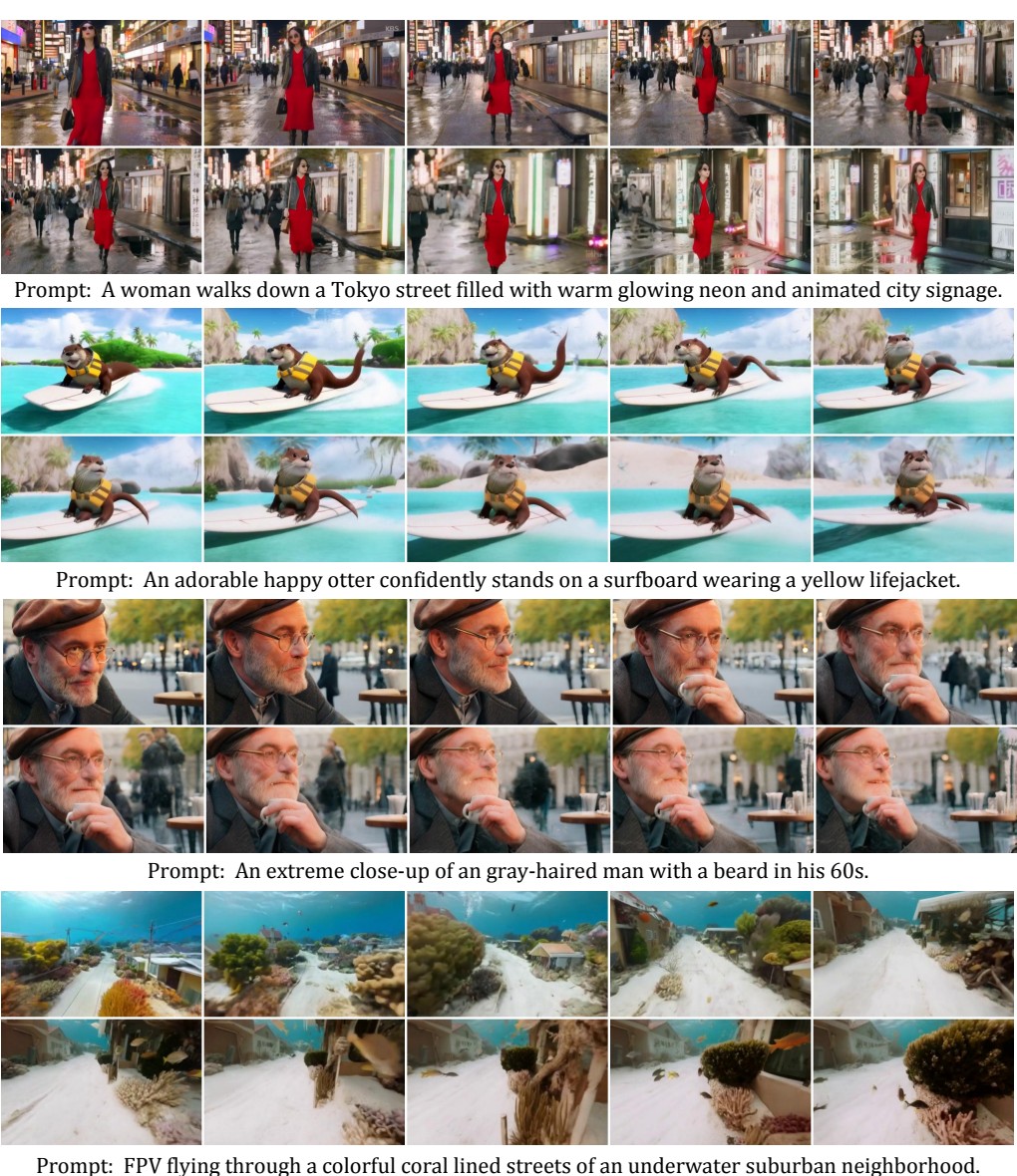

Prompt: A woman walks down a Tokyo street filled with warm glowing neon and animated city signage.

Prompt: An adorable happy otter confidently stands on a surfboard wearing a yellow lifejacket.

Prompt: An extreme close-up of an gray-haired man with a beard in his 60s.

Prompt: FPV flying through a colorful coral lined streets of an underwater suburban neighborhood.

Figure 9: **Zero-shot video extrapolation.** We extend the 4-second text-to-video result to 40 seconds.

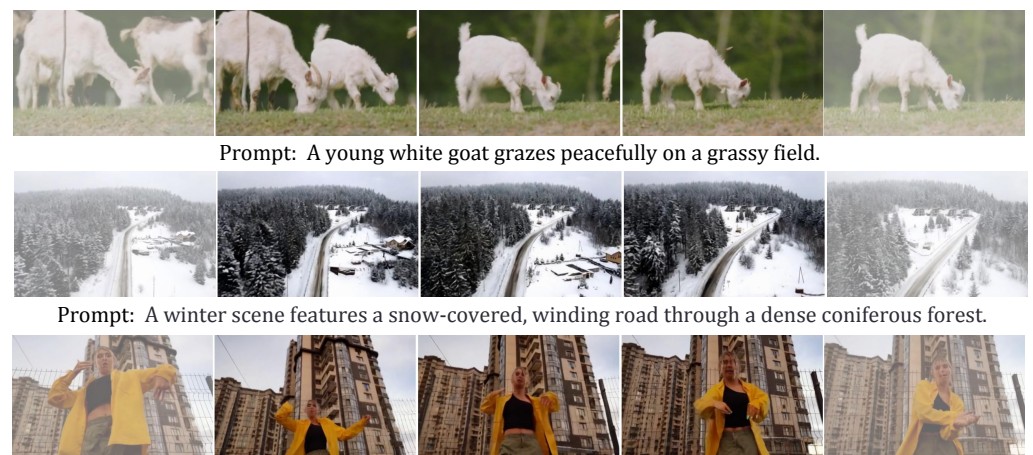

Prompt: A young white goat grazes peacefully on a grassy field.

Prompt: A winter scene features a snow-covered, winding road through a dense coniferous forest.

Prompt: A blonde dancer in a yellow raincoat performs an expressive routine on a rooftop.

Figure 10: **Zero-shot start-end frame control.** The start-end frames are rendered with transparency.