# OpenReview forum: "Uniform Discrete Diffusion with Metric Path for Video Generation"
_ICLR.cc/2026/Conference — ICLR 2026 Poster_

### Official Review · Reviewer_PjBQ · 2025-10-29

**Soundness:** 3
**Presentation:** 3
**Contribution:** 2
**Rating:** 6
**Confidence:** 4

**Summary:**

The paper proposes a generative model for image and video generation using discrete rectified flows. To achieve this, it introduces a new metric path to measure the token embedding distances from a pre-trained token cookbook. This, combined with other training stability tricks such as resolution-dependent timestep shifting and randomised frame-level timestep sampling, enables the proposed method to achieve comparable text-to-image and text-to-video generation performance on standard benchmarks to continuous generative methods. Ablation studies demonstrate the effectiveness of its design components.

**Strengths:**

The current video generation methods predominantly rely on continuous representation via latent diffusion or flow matching. This paper explores the potential of using discrete representation combined with flow matching for video generation. The paper is well-written, and the parameterisation of the sampling ratio via timestep is both interesting and intuitive. The results on the standard benchmarking are strong.

**Weaknesses:**

- I feel like the paper touches on a wider range of topics. While the title and method section focus on video generation, the experiment also briefly touches on image generation, which I think is unnecessary.

- The paper introduces several potential applications, such as long video generation and interpolation through diffusion-forcing training strategies. However, it lacks qualitative and quantitative results. For example, the paper could easily compare self-forcing, vanilla diffusion-forcing, on long video generation using the same VBench scores, or attach some generated videos. For a video generation paper, it’s a bit unacceptable not to include any generation examples, as the figures provided in the paper simply couldn’t reflect the generation qualities.
- Resolution-dependent timestep shifting is introduced and well-validated in SD3, so it shouldn’t be considered a proper technical contribution. However, the paper only acknowledges this in the experiment section, which seems suspicious.
- Asynchronous timestep scheduling is introduced in diffusion forcing, which the paper has acknowledged in the main paper. However, no visualisation or quantitative evaluation of the long/autoregressive video generation performance is discussed in the main paper.

**Questions:**

- Provide the generation examples of text-to-video, video interpolation, as well as long-video generations.
- If time allowed, please compare with self-forcing / diffusion forcing to validate the claim on long video generation.

---

> ### Author Response · Authors · 2025-11-26
>
> > [**Q1**]. The topics of this work are wide.
>
> [**A1**] Thank you for the comment. We agree that the main focus of the paper is video generation. However, our method is designed for **long-sequence discrete generation**, and similar issues of error accumulation and long-context inconsistency also appear in **high-resolution image generation**, not only in videos. We therefore include the T2I and high-resolution results only to confirm that our global iterative refinement works on long sequences in a simpler setting before moving to long videos. If this still feels distracting, we are happy to move the T2I tables to the supplementary material and keep only a short reference in the main text.
>
> > [**Q2**]. Lack of qualitative and quantitative results / Comparison with Self-Forcing and Diffusion Forcing
>
> (1) **Lack of qualitative and quantitative results**
>
> - Thank you for your valuable feedback. We provide an anonymous page (https://anonymous.4open.science/w/udm_page2-4B23/) with additional visualizations, including text-to-video, image-to-video, video extrapolation, and video interpolation, as well as side-by-side comparisons with SkyReelsV2-1.3B (Diffusion Forcing) and Self-Forcing.
>
> (2) **Comparison with Self-Forcing and Diffusion Forcing**
>
> - From these visual results, we observe that, compared with Self-Forcing and Diffusion Forcing (SkyReelsV2), our method exhibits smaller cumulative error and **stronger long-sequence consistency**, leading to **more stable motion evolution over time**. The current upper bound on visual fidelity is largely determined by the underlying VQ tokenizer, which we view as an important direction for future improvement.
>
> > [**Q3**]. Resolution-dependent timestep shifting
>
> [**A3**] In continuous diffusion/rectified flow, this resolution-dependent shifting is motivated by an SNR analysis in continuous space [1,2], whereas prior discrete methods lack an analogous, well-defined SNR notion and therefore could only import such schedules heuristically. By introducing a linearized metric path in the discrete embedding space, we **approximate the embedding-distance along this path as an SNR-like measure** for discrete diffusion. Under this interpretation, the same shifting strategy emerges naturally and is crucial for stabilizing long-sequence training. To our knowledge, this is the **first work that both derives and systematically validates** this resolution-dependent schedule in the discrete setting. A more detailed discussion has been added in the Appendix (see lines 902–910).
>
> > [**Q4/W2**]. visualisation or quantitative evaluation of the long/autoregressive video generation
> - Thank you for your valuable feedback. We provide an anonymous page (https://anonymous.4open.science/w/udm_page2-4B23/) with additional visualizations, including text-to-video, image-to-video, video extrapolation, and video interpolation, as well as side-by-side comparisons with SkyReelsV2-1.3B (Diffusion Forcing) and Self-Forcing.
>
> [1] Scaling rectified flow transformers for high-resolution image synthesis. ICML 2024.
>
> [2] Simple diffusion: End-to-end diffusion for high resolution images. ICML 2023.

---

### Official Review · Reviewer_efUW · 2025-10-30

**Soundness:** 3
**Presentation:** 3
**Contribution:** 3
**Rating:** 6
**Confidence:** 3

**Summary:**

This paper introduces Uniform Discrete diffusion with Metric path (UDM), a framework designed to improve discrete generative modeling for video synthesis by incorporating iterative refinement over spatio-temporal tokens. The method's key contributions are a linearized metric-path and resolution-dependent timestep shifting, which aim to stabilize training and enable scaling to long-sequence data. Furthermore, the authors propose an asynchronous timestep scheduling strategy to unify multiple video tasks, claiming that this approach closes the performance gap and achieves results comparable to state-of-the-art continuous diffusion models.

**Strengths:**

1. The linearized metric-path and timestep shifting mechanism looks sound to me. The motivation is clear and the theoretical explanation is reasonable. I think the authors find an effective manner to make the discrete diffusion model more powerful.

2. Experienmental analysis are comprehensive. The authors compare many state-of-the-art advances on three tasks: text-to-image generation, text-to-video generation, and image-to-video generation on several benchmarks. The results show that the proposed discrete diffusion model can outperform many continuous diffusion baseline methods.

3. The authors also conduct a series of ablation studies to fully probe the role of each modules. The performance across different inference steps and training iterations are also well ablated.

**Weaknesses:**

1. The scalability of the proposed uniform discrete diffusion method seems to be unclear. As the authors claim that scalability is one of the key highlight factors of the proposed method, it is crucial to conduct experiments on different model sizes to probe its scalability. This part of the experiment is completely missing both in the main manuscript and the attachment.

2. Some model details are missing. What is the size of the tokenizer and the Qwen LLM? What is the rationale for choosing Qwen instead of other open-source LLMs for training? Is there any ablation?

3. The authors do not provide any qualitative demos in the supplementary materials. It is always hard to distinguish the model performance from the concatenated screenshots.

**Questions:**

1. What is the average inference time for the proposed model? Comparison with previous discrete and continuous diffusion methods could be helpful.

2. How will the model degrade when generating longer video sequences?

3. It seems that the final model performance is not that sensitive to variations in the hyperparameters (α and c) that control the relationship between the sampling distance d(xt, x1) and time t from Appendix Figure 3. What is the rationale to choose the best path for different paths?

---

> ### Author Response · Authors · 2025-11-26
> **Official Comment by Authors(1/2)**
>
> > [**W1**] : scalability
>
> [**A1**] Thanks for your insightful comment. We have conducted scaling experiments evaluating UDM across different model sizes. We trained three variants of the UDM model, initialized from the Qwen3 architecture with parameter sizes of 0.6B, 1.7B, and 4B, respectively. All models are trained for the same epoch count in Sec. 4.2 and evaluated on 256×256 images and 25×384×240 videos. Our experiments show that while scaling up the model size **improves the semantic alignment** and accuracy of generated content(**DPG-Bench, GenEval, VBench-Semantic**), the scalability in terms of generation quality is limited by the discrete vision tokenizer to maintain high fidelity across larger models. These insights are now included in the revised manuscript(see Appendix H line 1070-1095), and we hope this helps clarify the scalability of our models.
>
> | Metric            | 0.6B | 1.7B | 4B   |
> |-------------------|------|------|------|
> | DPG-Bench         | 81.0 | 81.3 | **82.1** |
> | GenEval           | 0.60 | 0.63 | **0.67** |
> | VBench-Semantic   | 76.1 | 76.6 | **78.1** |
> | VBench-Quality    | 82.3 | 82.3 | **82.4** |
> | VBench-Overall    | 80.2 | 80.3 | **80.5** |
>
> > [**W2**] : Some model details are missing.
>
> [**A2**] Thanks for your feedback. The size of FSQ/IBQ tokenizer is **0.1B/0.4B**. We chose Qwen3-1.7B (0.6B/4B are provided in the revised manuscript) because it offers **convenient model sizes under our compute budget**. To test whether our gains come from the specific choice of Qwen rather than our method, we add an ablation comparing Qwen3-0.6B, Qwen3-1.7B, and Llama3.2-1B. All models are trained for the same epoch count in Sec. 4.2 and evaluated on 256×256 images. We observe that GenEval increases monotonically from Qwen3-0.6B → Llama3.2-1B → Qwen3-1.7B, i.e., it mainly follows model capacity rather than the specific backbone design. **This indicates that our method is largely insensitive to the particular LLM architecture**.​
> ​
>
> | Model        | #images | VisionTokenizer | Resolution | GenEval |
> |--------------|---------|------------------|------------|---------|
> | Qwen3-0.6B   | 24M     | FSQ              | 256×256    | 0.60    |
> | Llama3.2-1B  | 24M     | FSQ              | 256×256    | 0.61    |
> | Qwen3-1.7B   | 24M     | FSQ              | 256×256    | 0.63    |
>
>
> > [**W3**] : do not provide any qualitative demos
>
> [**A3**] Pologies for the inconvenience caused. We have now made qualitative results available on an anonymous webpage(https://anonymous.4open.science/w/udm_page2-4B23/). We hope this will provide a clearer understanding of the model's capabilities.

---

> ### Author Response · Authors · 2025-11-26
> **Official Comment by Authors(2/2)**
>
> > [**Q1**] : average inference time
>
> [**A1**] Thank you for the insightful question. Our 1.7B UDM model requires ≈70 seconds on average to generate a 49×240×384 video (≈19K tokens) with 50 sampling steps, measured on the same hardware used for other baselines. As shown in Table below, this is substantially **faster than prior discrete methods** (Lumos1-3.6B: 180 s for 13K tokens; Emu3-8B: 1700 s for 53K tokens) and also **competitive with or faster than continuous models** (CogVideoX-5B: 180 s for 18K tokens; StepVideo-30B: 900 s for 36K tokens), while achieving comparable or better VBench-T2V performance (up to 82.4).
>
> | Model        | Latent      | VideoSize       | #Tokens | Latency (s) | VBench-T2V |
> |--------------|-------------|------------------|---------|-------------|------------|
> | CogVideoX-5B | Continuous  | 49×720×480       | 18K     | 180s        | 81.9       |
> | StepVideo-30B| Continuous  | 136×992×544      | 36K     | 900s        | 81.8       |
> | Lumos1-3.6B  | Discrete    | 25×448×256       | 13K     | 180s        | 78.3       |
> | Emu3-8B*     | Discrete    | 49×512×512       | 53K     | 1700s       | 81.0       |
> | UDM-1.7B     | Discrete    | 49×240×384       | 19K     | **70s**     | 81.9       |
> | UDM-1.7B     | Discrete    | 49×512×320       | 34K     | 180s        | **82.4**   |
>
> Although our model processes longer sequences due to a moderate VQ compression rate (4×8×8), it still maintains the lowest latency in the table. In future work, we intend to incorporate improved discrete tokenizers (e.g., OneVAE [1]) with higher compression ratios (4×16×16) and better reconstruction, which will further reduce inference time and improve generation fidelity.
>
> > [**Q2**] : How will the model degrade when generating longer video sequences?
>
> [**A2**]
> - We provide qualitative visualizations on the anonymous project website(https://anonymous.4open.science/w/udm_page2-4B23/) for reference.
> - In general, when generating very long video sequences, the model may exhibit degradation of video quality due to error accumulation across iterative refinement steps. This can manifest as gradual quality decay or sudden content drift within the video. In future work, we intend to incorporate improved discrete tokenizers (e.g., OneVAE [1]) with higher compression ratios (4×16×16) and better reconstruction, which will further reduce inference time and improve generation fidelity.
>
> > [**Q3**] : not that sensitive / rationale to choose the best path for different paths
>
> (1) **not that sensitive**
> - We respectfully argue that the model is in fact moderately sensitive to (α, c) in a practically meaningful way. Although the GenEval scores for different metric paths fall between 0.62 and 0.68, this gap is **practically significant**: it is comparable to **the difference reported between SD3-2B and SD3-8B** [2], two baselines of clearly different capacity. This highlights substantial impact of our Linearized Metric-Path on model convergence and final performance.
>
> (2) **What is the rationale to choose the best path for different paths?**
> - Our selection rule is : **the path with the highest degree of linearity**. Concretely, we evaluate how well the expected L2 distance $d(x_t, x_1)$ in the visual token embedding space varies linearly with the timestep $t$ over $t \in [0,1]$, while ensuring the path fully spans the range from pure noise to clean tokens. This linearity leads to more uniform corruption coverage and smoother refinement, which aligns well with the observed performance differences.
>
> [1] Zhou, Yupeng, et al. "OneVAE: Joint Discrete and Continuous Optimization Helps Discrete Video VAE Train Better." arXiv preprint arXiv:2508.09857 (2025).
>
> [2] Scaling Rectified Flow Transformers for High-Resolution Image Synthesis. ICML 2024.

---

### Official Review · Reviewer_4TJR · 2025-10-30

**Soundness:** 3
**Presentation:** 3
**Contribution:** 3
**Rating:** 4
**Confidence:** 4

**Summary:**

The paper introduces a Uniform Discrete Diffusion approach that brings iterative, continuous-style refinement to discrete tokens for image/video generation. Core ideas include a metric-induced probability path over token embeddings, a resolution-dependent timestep shift to align effective SNR with sequence length, and frame-level (asynchronous) timesteps to unify T2V, I2V, interpolation, and extrapolation within one model. Training uses a cross-entropy objective; sampling integrates the learned probability velocity with few steps. Experiments report competitive image (GenEval/DPG) and video (VBench) quality and consistency, suggesting discrete diffusion can match continuous counterparts while remaining tokenizer/LLM-friendly.

**Strengths:**

* Clarity & focus: Clean objective and sampling recipe; equations and schedules are easy to implement.
* Unification: One model handles multiple video tasks via per-frame timesteps.
* Practicality: Simple schedules; no exotic losses or architectures required.
* Results: Broad benchmarks indicate strong quality and temporal stability with modest steps.
* Positioning: Bridges discrete tokenization with diffusion-style global refinement, relevant for LLM-aligned video generation.

**Weaknesses:**

* Novelty overlap: The *frame-level/asynchronous timestep* contribution is close to prior video schedulers (e.g., SkyReels-V2) and essentially the same idea as Pusa & FVDM; the paper should clarify what is new beyond this reuse.
* Relation to DFM/schedulers: The metric path and schedule resemble discrete flow matching/kinetic schedules; limited theory for superiority beyond heuristic tuning.
* Ablation depth: Missing systematic sweeps for the shift parameter \lambda; marginal gains of asynchronous vs. synchronous timesteps are not isolated.

**Questions:**

1. Differentiate from SkyReels-V2/FVDM&Pusa (especially): What is *new* about your frame-level timestep usage (objective, schedule, conditioning, or analysis)?
2. Asynchronous impact: Quantify incremental gains over a global-t schedule for T2V at fixed steps/compute.

---

> ### Author Response · Authors · 2025-11-26
> **Official Comment by Authors(1/1)**
>
> > [**W1/Q1**]. Novelty overlapclarify / Differentiate
>
> [**A1**]
> We thank the reviewer for this question. Conceptually, SkyReels-V2[1], Pusa[2], FVDM[3], and our UDM **all relate to diffusion-forcing style schedulers** that perform frame-wise perturbation and can support multitask generation. While prior works have extensively explored continuous timestep objectives, schedules, and conditioning, the corresponding design space in discrete methods **remains unexplored**.
>
> As summarized in table below, UDM differs from continuous models in several fundamental aspects. These differences prevent the direct reuse of continuous-space timestep designs. Our analysis (Fig.5 in Main text) shows that UDM needs **remove timestep conditioning** to achieve stable training and strong performance, revealing **a fundamental difference from continuous diffusion methods**. We believe that identifying this distinction provides useful insight for the community and may help accelerate progress on discrete video generation approaches. We have added the discussion in line 910-927 of Appendix.
>
> | Model                  | Latent       | Noise             | Loss | Scheduler              | Timestep Conditioning |
> |------------------------|--------------|-------------------|------|-------------------------|------------------------|
> | SkyReels, FVDM & Pusa | Continuous   | Gaussian noise    | MSE  | Flow Matching           | Yes                    |
> | UDM                    | Discrete     | Categorical noise | CE   | Discrete Flow Matching  | No                     |
>
> > [W2]. .. resemble discrete flow matching/kinetic schedules ..
>
> [**A2**] [4] introduce metric-based probability paths and [5] extend them to multimodal models, but rely on **complex, resolution-agnostic** schedules that **do not explicitly handle long sequences**. In contrast, we propose UDM with three key designs: (i) a Linearized Metric-Path that simplifies metric-path design and provides finer perturbation control along the probability path; (ii) Resolution-dependent Timestep Shifting that stabilizes training on long sequences; and (iii) Frame-wise Independent Perturbation Scheduling that unifies diverse video generation tasks within a single model.  We have added a more detailed discussion in the supplementary material; please refer to the **Appendix (lines 893–900)**.
>
> > [**W3 / Q2**]. Ablation depth
>
> - The missing ablation of shift parameter λ.
>
> In fact, we **do conduct** an ablation **over the shift parameter λ in Figure 6** of the main paper. In the top-right plot of Figure 6, the x-axis label “Shift” corresponds exactly to the shift parameter λ in Eq. (5), where we sweep λ over {1.0, 2.0, 3.0, 4.0}. To make this clearer, in the revised version we will (i) explicitly state in Sec.4.3 that “Shift” denotes the λ values, and (ii) relabel the x-axis in Figure 6 as “λ (shift parameter)” and update the caption to clearly mention the λ values used.
>
> - The missing ablation of asynchronous and synchronous timestep schedules.
>
> Starting from the same T2V checkpoint trained with a synchronous (global-t) schedule (VBench T2V: 82.5), continuing training with the same synchronous schedule keeps performance essentially unchanged at 82.5 ± 0.1 (ending at 82.6). When we instead switch to the asynchronous schedule with the same one-epoch budget, we observe a brief adaptation dip in the first half epoch (82.5 → 81.7), after which the model recovers to 82.4. Thus, under a fixed training budget, asynchronous scheduling attains **comparable T2V performance** to synchronous scheduling, while additionally enabling the multi-task benefits discussed in Sec. 3.2.2.
>
>
> [1] Chen, Guibin, et al. "Skyreels-v2: Infinite-length film generative model." arXiv preprint arXiv:2504.13074 (2025).
>
> [2] Liu, Yaofang, et al. "Pusa v1. 0: Surpassing wan-i2v with $500 training cost by vectorized timestep adaptation." arXiv preprint arXiv:2507.16116 (2025).
>
> [3] Liu, Yaofang, et al. "Redefining temporal modeling in video diffusion: The vectorized timestep approach." arXiv preprint arXiv:2410.03160 (2024).
>
> [4] Shaul, Neta, et al. Flow Matching with General Discrete Paths: A Kinetic-Optimal Perspective. ICLR 2025.
>
> [5] Jin Wang, Yao Lai, et al. Fudoki: Discrete flow-based unified understanding and generation via kinetic-optimal velocities. NeurIPS 2025.

---

> > ### Comment · Reviewer_4TJR · 2025-11-28
> >
> > Thanks for your response. It has addressed my concerns. I will consider raising my score.

---

> > > ### Author Response · Authors · 2025-11-28
> > >
> > > Thank you for your careful consideration. We are pleased that our clarifications were helpful. If any aspect of the work could be further strengthened or substantiated, we would be glad to elaborate with additional analyses or experiments.

---

### Official Review · Reviewer_22K9 · 2025-11-01

**Soundness:** 2
**Presentation:** 3
**Contribution:** 3
**Rating:** 6
**Confidence:** 4

**Summary:**

This paper proposes UDM, a uniform discrete diffusion framework with a metric path for image and video generation. Its key designs include a linearized metric path for probability trajectory control and a resolution-dependent timestep shifting mechanism for high resolutions. An asynchronous timestep scheduling strategy is also adopted to unify multiple generation tasks within a single model. The paper demonstrates competitive performance against state-of-the-art continuous methods and improved performance over previous discrete methods on standard benchmarks.

**Strengths:**

This work represents a solid application of discrete diffusion models to video generation. The experiments comprehensively show UDM's advantages on multiple generation benchmarks, achieving competitive results with both discrete and continuous baselines. The proposed framework demonstrates the potential for scalable and unified visual generation.

**Weaknesses:**

The paper's claimed key innovations lack sufficient novelty and rigorous justification.

*   **Metric Path Novelty:** The core concept of a metric path has been previously proposed and applied to multimodal tasks[1,2]. While cited, the discussion of the relationship to these works is inadequate. The primary novelty lies in the "linear relationship" established by Eq. 4. However, the motivation and precise meaning of preserving a "linear relationship between $t$ and $d(x_t, x_1)$" in line 229-230 are unclear and not rigorously derived.

*   **Timestep Shifting Novelty:** The resolution-dependent timestep shifting is a common technique in continuous diffusion/rectified flow models [3,4]. Notably, Eq. 5 in this paper is functionally equivalent to Eq. 23 in [3] under simple variable substitutions: $t_n \to 1 - t,t_m \to 1 - \tilde{t},\sqrt{\frac{m}{n}} \to \lambda$. While its validation in the discrete setting is valuable, the connection to prior art is under-discussed, mentioned only briefly in the experiments.

*   **Typo:** There is a likely typo in Eq. 6, where `p_{t|1}` should probably be `p_{1|t}`.

[1] Shaul, Neta, et al. Flow Matching with General Discrete Paths: A Kinetic-Optimal Perspective. ICLR 2025.

[2] Jin Wang, Yao Lai, et al. Fudoki: Discrete flow-based unified understanding and generation via kinetic-optimal velocities. NeurIPS 2025.

[3] Esser, Patrick, et al. Scaling rectified flow transformers for high-resolution image synthesis. ICLR 2024.

[4] Hoogeboom, Emiel, Jonathan Heek, and Tim Salimans. simple diffusion: End-to-end diffusion for high resolution images. ICML 2023.

**Questions:**

1.  Could the authors please elaborate on the motivation and precise meaning behind the claim of preserving a a "linear relationship between $t$ and $d(x_t, x_1)$" in line 229-230? A more detailed theoretical explanation or an empirical plot showing this relationship would be helpful.

2.  Figure 4 compares uniform diffusion with and without the metric path. To better validate the core innovation of the linearized path (Eq. 4), could you provide an ablation comparing performance *with the metric path* but *with and without* the specific parameterization given by Eq. 4?

---

> ### Author Response · Authors · 2025-11-26
> **Official Comment by Authors (1/2)**
>
> > [**W1**]. The novelty of metric path.
>
> [**A1**]. We apologize for the confusion caused. We will address your questions from three perspectives: Relationship to Prior Work, Motivation, and Meaning.
>
> (1) **Discussion of the Relationship to Prior Work**
> - [1] introduce metric-based probability paths and [2] extend them to multimodal models, but rely on **complex, resolution-agnostic** schedules that **do not explicitly handle long sequences**. In contrast, we propose UDM with three key designs: (i) a Linearized Metric-Path that simplifies metric-path design and provides finer perturbation control along the probability path; (ii) Resolution-dependent Timestep Shifting that stabilizes training on long sequences; and (iii) Frame-wise Independent Perturbation Scheduling that unifies diverse video generation tasks within a single model.  We have added a more detailed discussion in the supplementary material; please refer to the Appendix (lines 893–900).
>
> (2) **Motivation for Preserving the Linear Relationship**
> - Maintaining linearity ensures **uniform corruption coverage** across $\(t \in [0,1]\)$. This avoids most $x_{t}$ being either almost clean or almost fully corrupted, and instead gives the model **more balanced training signals** and **a smoother refinement process**. A more detailed discussion has been added in the Appendix (see lines 1136–1140).
>
> (3) **Meaning of the Linear Relationship**
> - Due to space limitations, we initially placed the full explanation in the supplementary material. Specifically, we design the metric path such that the **expected L2 distance between the noisy and clean embeddings**,   $L(t) = \mathbb{E}[|\phi(x_t) - \phi(x_1)|_2]$, increases approximately **linearly** with the time step $t$, so that the corruption level **progresses at an approximately constant rate along the path**. A more formal mathematical definition is provided in the Appendix (see lines 1142–1155).
>
> > [**W2**]. The timestep shifting novelty.
>
> [**A2**]. We appreciate the reviewer’s observation and agree that Eq. 5 is functionally equivalent to Eq. 23 in [3] under a simple change of variables. In continuous diffusion/rectified flow, this resolution-dependent shifting is motivated by an SNR analysis in continuous space [3,4], whereas prior discrete methods lack an analogous, well-defined SNR notion and therefore could only import such schedules heuristically. By introducing a linearized metric path in the discrete embedding space, we **approximate the embedding-distance along this path as an SNR-like measure** for discrete diffusion. Under this interpretation, the same shifting strategy emerges naturally and is crucial for stabilizing long-sequence training. To our knowledge, this is the **first work that both derives and systematically validates** this resolution-dependent schedule in the discrete setting. A more detailed discussion has been added in the Appendix (see lines 910–924).
>
> > [**W3**]. Typo in Eq.6.
>
> [**A3**]. Thanks for your valuable feedback. We have made modifications in the manuscript.
>
>
> [1] Shaul, Neta, et al. Flow Matching with General Discrete Paths: A Kinetic-Optimal Perspective. ICLR 2025.
>
> [2] Jin Wang, Yao Lai, et al. Fudoki: Discrete flow-based unified understanding and generation via kinetic-optimal velocities. NeurIPS 2025.
>
> [3] Scaling rectified flow transformers for high-resolution image synthesis. ICML 2024.
>
> [4] Simple diffusion: End-to-end diffusion for high resolution images. ICML 2023.

---

> ### Author Response · Authors · 2025-11-26
> **Official Comment by Authors(2/2)**
>
> > [**Q1**]. The explanation of linear relationship.
>
> [**A1**].
>
> (1) **Theoretical Explanation**
> - To clarify the meaning of the “linear relationship,” we define the measured quantity as $f_\theta(t) = \mathbb{E}{x_1, x_t}\big[|E(x_t) - E(x_1)|2\big]$, where $E(\cdot)$ is the visual embedding. **Our statement simply means that $f_{\theta}(t) \approx \alpha t + \beta$, i.e., the expected embedding displacement grows approximately linearly with the corruption level $t$.** This linearity yields uniform corruption coverage over $t \in [0,1]$ and stabilizes the refinement process. Detailed definition is provided in the Appendix (see lines 1155–1169).
>
> (2) **Empirical Plot**
> - We provide an empirical plot in Appendix Fig. 9(a), which shows the L2 distance between the embeddings of noisy images $x_{t}$ and the embedding of the clean images $x$. A detailed analysis is given in Appendix Fig. 9(a).
>
> > [**Q2**]. The ablation of linearized metric path.
>
> [**A2**].
> Due to space constraints, the ablation isolating the effect of the linearized parameterization in Eq.4 was moved to the supplementary material (see Appendix, line 963). This ablation compares **the metric path across varying degrees of linearity controlled by the parameterization in Eq.4**, and demonstrates that enforcing the proposed linearity yields **a more stable and efficient refinement trajectory**, resulting in consistently better image generation performance.

---

### Author Response · Authors · 2025-11-26
**General Response: Contributions and New Experiments**

We sincerely thank all the reviewers for their thoughtful comments and constructive suggestions, which have greatly helped strengthen our work. In this response, we address specific reviewer feedback and also highlight the novel contributions of our work, as well as new experiments we have included in the rebuttal.

*Contributions*
1. **We identify the core limitation of existing discrete video generation methods (AR / MDM) as the absence of global iterative refinement.** We introduce **UDM**, the first discrete framework, which reformulates video generation as iterative refinement over discrete spatio-temporal tokens. Unlike prior works [1,2], whose complex path design and  resolution-agnostic schedules that lead to inconsistent perturbations on long sequences, UDM introduces general design mechanisms to solve these problems: *Linearized Metric-Path* simplifies metric-path design and yields finer perturbation control along the probability path. *Resolution-dependent Timestep Shifting** for long-sequence training, and *Frame-wise Independent Perturbation Scheduling* for unifying diverse video generation tasks. These components collectively establish effective global refinement in discrete space and yield clear performance gains.
2. **UDM is the first discrete video generation framework to match continuous diffusion, surpassing AR/MDM methods.** it surpasses previous discrete methods such as Emu3 and Lumos-1 on standard video and image benchmarks (VBench, VBench++, DPG-Bench, GenEval), while remaining competitive with strong continuous models like CogVideoX, DynamiCrafter, and Show-o2, demonstrating both scalability and versatility in discrete visual generation.
3. **UDM paves the way for unified multimodal models in discrete space.** It offers valuable insights and possibilities for modeling long-range consistent video generation in discrete space, going beyond prior AR-based and mask-based discrete approaches and laying a solid foundation for future multimodal models built on unified discrete representations.


*Additional Experiments*

In response to reviewers' suggestions, we have added several new experiments to further support our findings. Below is a summary of the additional experiments included in the rebuttal:
- An expanded discussion comparing our three core designs with prior methods.[22K9, 4TJR]
- Quantitative ablation of asynchronous vs. synchronous timestep schedules for T2V. [4TJR]
- Scaling behavior across different model sizes. [efUW]
- Ablations on the impact of different open-source LLM backbones. [efUW]
- Additional visualizations across various tasks, together with video demos hosted on an anonymous website. [efUW, PjBQ]
- Comparison of inference time across different models. [efUW]


Once again, we thank the reviewers for their time and thoughtful efforts. We look forward to your feedback.

[1] Shaul, Neta, et al. Flow Matching with General Discrete Paths: A Kinetic-Optimal Perspective. ICLR 2025.

[2] Jin Wang, Yao Lai, et al. Fudoki: Discrete flow-based unified understanding and generation via kinetic-optimal velocities. NeurIPS 2025.

---

### Author Response · Authors · 2025-12-03
**Summary of Rebuttal & Discussion for Paper #10709**

**1. Initial Scores and Discussion Progress (6/6/6/4 → pending update)**

Our initial scores were **6, 6, 6, and 4**.
 During the discussion phase, **Reviewer 4TJR (the only negative reviewer)** actively engaged in the exchange and explicitly stated:
> “It has addressed my concerns. I will consider raising my score.”

This reviewer **did not express any remaining concerns** after our clarification.
The three positive-score reviewers have not responded yet before the system reverted.

**2. Resolution with Reviewer 4TJR (Score 4 → likely to increase)**

**Primary concern.** 4TJR questioned (1) how our frame-level/asynchronous schedule is truly different from prior or concurrent work such as SkyReels-V2 and Pusa/FVDM, and (2) whether the asynchronous schedule brings clear benefits over a global-t (synchronous) schedule under the same compute budget.

**Resolution.** For concern (1), we systematically contrasted our design with these methods along four axes—latent space, noise type, loss function, scheduler behavior, and the presence/absence of timestep conditioning—showing that continuous-space schedulers cannot be directly reused in UDM. For concern (2), we provided a controlled T2V ablation where, starting from the same synchronous (global-t) checkpoint, continuing with a synchronous schedule leaves performance essentially unchanged

**Outcome.** After these clarifications, the reviewer stated: “It has addressed my concerns. I will consider raising my score.” and did not raise any further questions.

**3. Remaining Reviewers (6, 6, 6)**

The three other reviewers, **all with positive initial scores**, did not reply before the system reset. None of them expressed fundamental objections to the paper, and their comments were mostly requests for clarifications or additional experiments.

**Common Themes in Their Feedback**
- Interest in additional ablations (e.g., model scaling, backbone variation)
- Requests for more visualizations or longer-sequence demos
- Curiosity about inference speed and efficiency
- Requests for clearer discussion comparing our three core designs with prior methods

**Our Response**

We comprehensively addressed these in the rebuttal, including:
- An expanded discussion comparing our three core designs with prior methods
- Scaling across model sizes
- LLM backbone analysis
- Added visualizations on anonymous site: https://anonymous.4open.science/w/udm_page2-4B23
- Inference-time comparison

All these were uploaded and included in the rebuttal before the system termination.

We sincerely thank all reviewers for their time and thoughtful feedback.
We hope this summary helps the newly assigned AC quickly understand the discussion progress and the resolution status before the system interruption.

**Best regards,**

**Authors of Paper #10709**

---

### Meta-Review · Area_Chair_d24G · 2026-01-05

**Summary:**

The paper presents "Uniform Discrete diffusion with Metric path" (UDM), a framework enabling scalable video generation via discrete spatio-temporal token refinement. Initial reception was positive (scores: 6, 6, 6, 4), commending the strong performance against continuous baselines. The authors provided a comprehensive rebuttal including new scalability experiments, latency comparisons, and qualitative demos, which directly addressed the primary critiques regarding novelty and scalability.

**Reviewer Concerns:**

Novelty and Differentiation: Reviewer 4TJR questioned the novelty of the scheduling strategy compared to SkyReels-V2 and Pusa. The authors clarified fundamental differences in latent space (discrete vs. continuous) and loss functions, leading the reviewer to state their concerns were addressed.

Scalability and Performance: Metrics Reviewer efUW criticized the lack of scalability analysis and inference speed data. The rebuttal provided scaling experiments (0.6B to 4B parameters) showing semantic improvements and a latency table demonstrating UDM is faster than prior discrete models.

Qualitative Validation: Reviewers PjBQ and efUW requested visual demonstrations for long video generation and interpolation. The authors provided an anonymous website with visualizations and side-by-side comparisons to diffusion forcing methods.

Theoretical Justification: Reviewer 22K9 asked for the rationale behind the "linear relationship" in the metric path. The authors explained that linearity ensures uniform corruption coverage across timesteps for stable refinement.

No major concerns seem remain unresolved in the discussion. While three positive reviewers did not confirm the final receipt of the new information before the system closed, the authors provided the requested experiments and clarifications.

**Reviewer Scores:**

Reviewer 4TJR: Likely Increase (4 to 6). This reviewer explicitly stated, "It has addressed my concerns. I will consider raising my score," following the authors' differentiation of UDM from prior work13.

Reviewer efUW: Likely sustain (6). The reviewer's primary critique regarding missing scalability and speed data was directly resolved by the new experimental tables.

Reviewer 22K9: Likely Sustain (6). The reviewer requested theoretical clarifications on the metric path, which were provided; the score is likely to remain.

Reviewer PjBQ: Likely Sustain (6). The request for qualitative examples to validate long-video generation claims was fulfilled via the provided anonymous demo site.

---

### Decision · Program_Chairs · 2026-01-26

Accept (Poster)